# Sleep Problems in Pregnancy—A Cross-Sectional Study in over 7000 Pregnant Women in Poland

**DOI:** 10.3390/ijerph17155306

**Published:** 2020-07-23

**Authors:** Magdalena Smyka, Katarzyna Kosińska-Kaczyńska, Nicole Sochacki-Wójcicka, Magdalena Zgliczyńska, Mirosław Wielgoś

**Affiliations:** 1Department of Obstetrics and Gynecology, Medical University of Warsaw, Starynkiewicza Square 1/3, 02-015 Warsaw, Poland; magda.smyka@gmail.com (M.S.); nicole.sochacki-wojcicka@wum.edu.pl (N.S.-W.); miroslaw.wielgos@wum.edu.pl (M.W.); 2Department of Obstetrics and Gynecology, The Center of Postgraduate Medical Education, Cegłowska St. 80, 01-809 Warsaw, Poland; zgliczynska.magda@gmail.com

**Keywords:** sleep disorders, pregnancy, insomnia, epidemiology, sleep, pregnant women

## Abstract

The aim was to characterize sleep patterns in pregnant women in Poland and to analyze the relation between sociodemographic factors, pregnancy-related physical symptoms and sleep problems. A self-composed questionnaire, containing questions in Polish language, was distributed online via web pages and Facebook groups designed for pregnant women. The questionnaire included questions regarding sociodemographic data and information on the current pregnancy and sleep patterns over the past four weeks. Exactly 7207 respondents were included to the study. 77.09% reported sleep problems: nocturnal awakening (52.77%), sleep onset insomnia (20.23%), awakening too early (18.56%) and believed their sleep was too shallow (9.82%). Sleep onset insomnia (26.38%) and frequent awakening (62.88%) were most often reported in 3rd trimester, while daytime drowsiness (86.35%) and naps impeding daytime function (27.18%) in the 1st trimester of pregnancy. The analyzed demographic, socioeconomic and psychological issues had a small but significant influence on sleep problems occurrence (aOR 0.81–1.24). Time to conceive from 6 to 12 months of trying increased the risk of developing sleep problems during pregnancy (aOR 1.31). Pregnancy ailments increased the risk of sleep disturbances (aOR 1.53–2.59). Sleep disorders are prevalent among pregnant women in Poland. The evaluation of risk factors is essential in proper screening for sleep disorders in pregnant women.

## 1. Introduction

Sleep problems are common in the majority of pregnant women. Physical, psychological and hormonal changes make pregnant women more likely to suffer from sleep disturbances. The most common sleep-related disorders during pregnancy include insomnia, restless legs syndrome, obstructive sleep apnea and night-time gastroesophageal reflux disease. Available studies estimated the occurrence of sleep disorders in pregnant women at 46–78%, with the quality of sleep declining towards the third trimester [1,2,3,4]. Almost 80% of pregnant women suffer from insomnia in the third trimester of pregnancy [5,6]. According to the literature the occurrence of insomnia in the Polish population of pregnant women was estimated at 25–40%, while overall sleep disorder rate at 84.2–90.5% [7,8,9]. The wide range of reported occurrence is due to methodological differences of the surveys and not numerous study groups. No large study on pregnancy sleep quality in pregnant women in central Europe has been published till date.

Some authors reported that sleep disorders are a risk factor of gestational diabetes mellitus, hypertension, preeclampsia or intrauterine growth restriction and preterm delivery [10,11]. Pregnancy insomnia may also increase the risk of postpartum depression [12]. As the consequences of poor sleep are well known, it is crucial to establish the risk factors of sleep problems during pregnancy.

The aim of the survey was to characterize sleep patterns in pregnant women in Poland according to their subjective opinion and to analyze the relation between sociodemographic factors, pregnancy-related physical symptoms and sleep disorders.

## 2. Materials and Methods

A cross-sectional survey was conducted. A self-composed questionnaire, containing questions in Polish language, was distributed via internet between February and March 2018. It was composed of questions derived from several validated questionnaires (PSQI, Insomnia Severity Index, Stanford Sleep Questionnaire and Berlin Questionnaire). The questionnaire was administered by Google Forms (Google LLC, Mountain View, California, US). It was dedicated to pregnant Polish speaking women, regardless of inhabitancy and was distributed by web pages and Facebook groups, which were designed for pregnant women. Web pages written in Polish language provided general medical and social information on pregnancy and were available without any limits. Link to the questionnaire was presented also on Facebook groups bringing together women declaring being pregnant at the time. Facebook groups were accessible for women declaring being pregnant without any special requirements. Neither Facebook groups nor web pages were advertised at the time of the survey. On websites and Facebook group a short information on sleep problems in pregnancy and its importance was presented with with an invitation to complete the survey. No specific criteria, other than singleton pregnancy at the time of the survey were used for recruitment for the study. The survey was voluntary and anonymous—no questions regarding personal data that would enable the identification of participants were included and only the authors of the study had access to the collected information. Access to the questionnaire was granted after login in only once.

The first part of the questionnaire included questions regarding sociodemographic data and information on the current pregnancy, including pregnancy ailments. The second part contained questions regarding sleep patterns over the past four weeks. The questions regarded subjective sleep quality assessment, sleep problems including the occurrence of sleep onset insomnia (problems with falling asleep three nights a week or more, lasting at least one month and leading to the worsening of daytime functioning), awakening too early (in a subjective opinion), frequent nocturnal awakenings (at least two per night), sleep duration, the use of sleep medications and increased daytime drowsiness or naps impeding daytime function. The respondents were classified as declaring sleep problems if they reported one of the disturbances: sleep onset insomnia, frequent awakening, awakening too early in the morning or a belief that the sleep was too shallow. Questions regarding sharing a bed or a room with another person while sleeping were included in the questionnaire. For the assessment of life stress level and satisfaction with the current relationship a visual numerical scale from 0 to 10 was used. The questionnaire was composed by the authors and not validated.

Women aged over 18 years old and pregnant at the time of the survey were included in the study. 7855 respondents were included. Only questionnaires which were filled in completely were considered in the analysis. Women in multiple pregnancies were excluded from the study. Women declaring having sleep problems before the pregnancy were also included in the analysis. The answers were double-checked by the researchers and there were no identical records.

A power analysis was performed to assess the minimal number of respondents. For the 90% power, with a value of the zero hypothesis 90% (as basing on the available published polish data we assumed that 90% of pregnant women declare having any sleep problems), 1000 women should be included in the study (G*Power 3.1.9.4, Franz Faul, Christian-Albrecht University, Kiel, Germany).

Data were reported as absolute numbers and percentages or means and standard deviations. Statistical analyses were performed using Statistica version13.3 (Tibco Software Inc., Palo Alto, CA, USA). The Mann–Whitney test or Fisher exact tests were used to compare the analyzed variables. All tests were two tailed and *p* < 0.01 was considered significant. Logistic regression analysis was performed to assess which issues were independent risk factors of sleep problems during pregnancy.

The study protocol obtained the approval of the Ethics Committee of the Medical University of Warsaw (no 124/2018). The committee waived the obligation to obtain a written or verbal consent to participate in the study as completing the questionnaire was deemed tantamount to giving consent.

## 3. Results

A total of 7855 pregnant women took part in the survey. As 384,426 women delivered in 2018 in Poland, women included in the study would account for 2% of the whole population. After excluding questionnaires filled in incompletely, 7207 respondents were included into the study. The basic characteristics of the study group are presented in Table 1. A total of 791 (10.96%) respondents were in the first, 2580 (35.8%) in the second and 3836 (46.76%) in the third trimester of pregnancy.

Reported sleep variables in the whole study group and in pregnancy trimesters are presented in Table 2.

The study group was further divided into women declaring having or not at least one of the sleep problems listed in the questionnaire. A total of 5556 (77.09%) respondents reported sleep problems, while 1651 (22.91%) declared not having ones. Sleep problems were reported by 71.8% of women in the first trimester, 73.6% in the second trimester and 80.6% in the third trimester of pregnancy. 20.23% of women declaring sleep disorders reported their main problem to be sleep onset insomnia, 18.56% awakening too early, and 9.82% believed their sleep was too shallow. The most common sleep disorder was nocturnal awakening, reported by 52.77% of women. Other reported problems were difficulties with waking up (7.32%), night-time anxiety (2.17%) and having nightmares (3.19%). 125 women admitted to using sleep medications during pregnancy (1.73%). Drowsiness during daytime was declared by 69.04% of the respondents reporting sleep problems: 86.35% in the first, 54.76% in the second and 77.05% in the third trimester of pregnancy. On the other hand only 47.1% of women declaring not having sleep problems reported excessive sleepiness (*p* < 0.01). A total of 70.2% of women with sleep problems admitted to taking one and 5.1%—at least two naps during the day. However, no differences in the frequency or duration of naps were found between women with and without sleep disorders. No differences in the number of women sharing a bed, sharing a room or sleeping alone in a separate room were also observed. Sleep variables of women in each trimester of pregnancy are presented in Table 2.

Characteristics of the respondents with and without sleep problems are presented in Table 1.

Women reporting sleep problems were older (mean 28.19 years old vs. 27.78 years old, respectively, *p* < 0.01), they were less frequently unemployed or studying (16.42% vs. 20.16%, *p* < 0.01) and more commonly did physical work for a living (18.92% vs. 16.23%, *p* = 0.01). They assessed their economic status as bad or sufficient (16.97% vs. 13.45%, *p* = 0.001) and rated their life stress level as higher (mean 4.76 points vs. 4.16 points, *p* < 0.001). Women declaring sleep problems assessed the satisfaction with the current relationship as lower (mean 8.77 points vs. 9.02 points, *p* < 0.001). They less often conceived during the first six months of trying (55.42% vs. 60.87%, *p* < 0.001) and more often during the second six months of trying (14.47% vs. 10.66%, *p* < 0.001).

The correlation between pregnancy-related physical symptoms and reported sleep problems was analyzed. The respondents declaring sleep problems more commonly suffered from nausea (12.2% vs. 6.1%; *p* < 0.001), vomiting (4% vs. 1.4%; *p* < 0.001), back pains (43.1% vs. 22.4%; *p* < 0.001), frequent nocturnal urination (77.7% vs. 55.4%; *p* < 0.001), breathing difficulties in supine position (24.9% vs. 7.8%; *p* < 0.001), leg cramps (33.4% vs. 18.8%; *p* < 0.001) and fetal movements causing sleep problems (24.2% vs. 7.8%; *p* < 0.001).

Logistic regression analysis was performed to identify which analyzed factors affected the risk of sleep problems. All factors with *p* < 0.01 in the above analyses were taken into analysis but only those who were found independent risk factors are presented in Table 3.

Several factors were found to have an independent effect on the occurrence of sleep problems. Age, bad or sufficient economic status, and high stress level were associated with increased risk of sleep problems, while high assessment of relationship and unemployment were associated with decreased occurrence of sleep problems in pregnancy. Time to conceive from 6 to 12 months of trying increased the risk of developing sleep problems during pregnancy (aOR1.31). A stronger relationship was observed between pregnancy ailments and sleep problems. Leg cramps increased the risk of sleep disturbances 1.5-fold, while all other complaints -twice or more. Breathing difficulties in supine position exerted the strongest impact on sleep problem occurrence (aOR 2.59).

## 4. Discussion

The present study demonstrated that 77% of pregnant women in Poland suffered from sleep disorders. The occurrence of those problems increased during pregnancy, with the highest incidence in the third trimester. Our results stay in line with Polish data published by other researchers, however in much smaller samples. In a study by Wołyńczyk–Gmaj et al. a group of 266 women in the third trimester (84.2%) reported various sleep problems. 59% of them suffered from night-time awakening, 24.8% from non-restorative sleep, 23.3% had problems falling asleep and 20.7% reported early morning awakenings [7]. Similarly, our study showed that more than half of the respondents reported nocturnal awakenings. Skoczylas et al. reported a high incidence of sleep disorders in the third trimester of pregnancy [9]. The authors distributed a questionnaire in a cohort of 147 patients hospitalized for labor between 38 and 41 weeks of gestation and found an incidence of reported sleep problems of 94.3%. Our results are similar to data published by other authors worldwide and may be an important voice in worldwide analysis of sleep during pregnancy. A high prevalence of sleep disorders during pregnancy is reported in all published research. The US National Sleep Foundation’s Women and Sleep Survey conducted in 1998 revealed that 78% of women reported disturbed sleep during pregnancy [1]. In Chinese the overall occurrence of sleep disorder-related symptoms was 56.1% [13]. The most common problems were daytime sleepiness (52.6%), nocturnal arousal (46.5%), insomnia (35.1%) and snoring (30.2%). According to Lopez et al. the highest occurrence of insomnia in the Brazilian population was reported in the first trimester (64%) and in the second trimester (70%), and lower in the third trimester of pregnancy (39%). Pregnant women more commonly suffered from excessive daytime sleepiness with the highest rate of 55% in the second trimester and specific awakenings with the highest rate of 84% in the third trimester [14]. It is an interesting finding as excessive sleepiness is a common complaint in the first trimester of pregnancy, however in our study it was reported mostly by women in the third trimester. Further analysis of sleep problems of pregnant women of different races is required. Dorheim et al. estimated the prevalence of insomnia in a group of 2816 pregnant Norwegian women at 61.9% [15]. According to Kızılırmak et al. 52.2% of Turkish women suffered from sleep disturbances during pregnancy [16].

In our study the occurrence of reported sleep disturbances increased with the duration of pregnancy. The increasing incidence with the highest rate in the third trimester was confirmed by Hashimi et al. and Sedov et al. [1,4]. According to Kızılırmak et al. the risk of insomnia in the third trimester of pregnancy was 2.03-fold higher than in the first and second trimester and 2.19-fold higher for women at the age of 20 and more than in younger ones [16]. In our study women’s age was also an independent risk factor of sleep disturbances during gestation. Some authors claimed an increased body mass index to be a risk factor of sleep disorders [13,17,18]. In the present study no such relation was found.

The present results showed that sleep problems were frequently associated with night-time urination, pain (specified as back pain), breathing problems, and leg cramps or fetal movements. Similar pregnancy aliments related to sleep problems were reported by Skoczylas et al. The authors found night-time urination (55%) and fetal movements (34%) to be the most frequent ones. Other commonly reported problems were stress (42%), uncomfortable position (68%) or GERD symptoms (34%) [9]. Hutchison et al. estimated frequent urination, breathing problems, pain and uncomfortable position to be the most significant factors influencing sleep (96%; 23%; 35.8%; 67.1%, respectively) [19]. Kızılırmak et al. found that the most common reasons of insomnia were frequent visits to the toilet (39.3%), not finding a comfortable position while sleeping (30.7%), and restless legs (13.7%). The authors found a relationship between depressive symptoms and a risk of developing insomnia during gestation. Women suffering from depression had a 2.63-foldhigher risk of sleep disorders than those who did not have depressive symptoms [16]. A similar association between depressive symptoms and sleep disorders was confirmed by other studies. According to Field et al. depressed women more often experienced sleep disturbances and higher depression, anxiety and anger scores during both the second and third trimesters [20]. Okun et al. found that at 20 and 30 weeks of gestation sleep was more disturbed in depressed compared to non-depressed pregnant women [21].

Our study revealed that 7–8 h of sleep was the most common sleep duration reported by pregnant respondents. A detailed analysis of sleep duration in each trimester of pregnancy in a large cohort of pregnant women is unique. According to Kızılırmak et al. the average sleep duration reported by pregnant women was 7 ± 2.26 h (first trimester: 8.3 ± 2.0, second trimester: 8.1 ± 2.1, third trimester: 7.3 ± 2.3) [16]. A similar result was reported by Facco et al. (mean 7.4 h) and Noudet al. (mean 7.2 h) [17,22]. Cai et al. assessed sleep duration only in the third trimester but it was also similar to previously reported results (8.0 ± 1.3 h) [13].

The strength of our study is a uniquely large cohort of pregnant women. The anonymity and distribution of the questionnaire via the internet might have promoted the honesty of the answers. To our knowledge no other research in such a large pregnant woman cohort in Poland has been conducted and published to date. However, there are some limitations of the study. The analyzed data are derived from a self-composed questionnaire, which could be the cause of an inherent bias. It was distributed online, therefore the sample may be biased from only those who would respond to an on-line survey. The question of reliability of the results is a valid concern. Information on employment was not detailed, without distinction between full-time and part-time or shift-job, which could induce bias to the results. However, the internet was chosen to administer this survey due to its ubiquitous nature. Despite its obvious limitations as a research tool (like impossibility to obtain verifiable information, as the survey enables to collects only objective opinions of the respondents and cannot rule out lies), it allows to reach a much larger and more diverse group of people all over the country. However, it allows to included only women using this kind of media. As only women pregnant at the time of the survey were included, no control group of non-pregnant women was available in the analysis. The lack of validated objective and subjective tests is another limitation of the study, as well as the subjective assessment of sleep problems by the respondents. A questionnaire is not an objective tool for the diagnosis of any sleep disorders. Conversely, choosing an objective and validated diagnosing tool as polysomnography would result in a significant reduction in the size of the study group.

## 5. Conclusions

Sleep problems are prevalent among pregnant women in Poland and increase significantly as the pregnancy progresses. Demographic, socioeconomic and psychological issues had a small but significant influence on sleep problems occurrence and pregnancy ailments significantly increased the risk of sleep disturbances. The evaluation of risk factors is essential in proper screening for sleep disorders in pregnant women and sleep hygiene could be discussed prior to pregnancy.

## Figures and Tables

**Table 1 ijerph-17-05306-t001:** Characteristics of the study group and respondents declaring having or not having sleep problems.

		Study Group *N* = 7207		Sleep Problems *N* = 5556		No Sleep Problems *N* = 1651		
		Number	%	Number	%	Number	%	*p*
Age (years) *^,Ώ^		28	3.8	28.19	3.83	27.78	3.87	<0.01
Pre-pregnancy BMI (kg/m^2^) *^,Ώ^		23.25	4.3	23.3	4.29	23.08	4.18	0.07
Primiparity ^Ώ^		4954	68.74	3767	67.8	1187	71.89	0.002
Pregnancy trimester ^#^	First	791	10.96	568	10.22	223	13.51	<0.001
	Second	2580	35.8	1898	34.16	682	41.31	<0.001
	Third	3836	46.76	3090	55.62	746	45.18	<0.001
Education ^#^	Primary	35	0.49	28	0.05	7	0.42	0.8
	Vocational	147	2.04	105	1.89	42	2.54	0.1
	Secondary	1925	26.71	1487	27.22	438	26.54	0.5
	Higher	5100	70.76	3936	70.84	1164	70.5	0.8
Inhabitancy ^#^	Village	1708	23.7	1307	23.52	401	24.29	0.5
	<50,000 inh.	1426	19.79	1094	19.7	332	20.11	0.7
	50,000–200,000 inh.	1598	22.17	1252	22.53	346	20.97	0.18
	>200,000 inh.	2475	34.34	1903	34.25	572	34.63	0.77
Employment ^#^	Unemployed	1246	17.3	913	16.43	333	20.17	<0.001
	Physical work	1319	18.25	1051	18.92	268	16.23	0.01
	Non-physical work	4642	64.45	3592	64.65	1050	63.6	0.4
Marital status ^#^	Single	44	0,61	36	0.66	8	0.49	0.6
	In relationship	1578	21.89	1185	21.32	393	23.8	0.04
	Married	5585	77.5	4335	78.02	1250	75.71	0.05
Time to conceive ^#^	Unplanned conception	1521	21.1	1178	21.2	343	20.78	0.7
	<6 months	4084	56.67	3079	55.42	1005	60.87	<0.001
	6–12 months	934	13	804	14.47	176	10.66	<0.001
	>12 months + ART	668	9.23	495	8.91	127	7.69	0.13
Stress level (points) *^,Ώ^		4.62	0.82	4.76	1.92	4.16	1.91	<0.001
Assessment of relationship (points) *^,Ώ^		8.82	2.48	8.77	1.57	9.02	1.41	<0.001

* mean/±SD; ^#^ Mann–Whitney test; ^Ώ^ Fisher exact tests; inh.: inhabitants; BMI: body mass index; ART: assisted reproduction techniques; descriptive statistics for the study group: Age: median 29, min. 17, max. 44, mode 27; pre-pregnancy BMI: median 22, min. 15, max. 51, mode 21; stress level: median 5, min. 1, max. 10, mode 2; assessment of the relationship: median 8, min. 1, max. 10, mode 8.

**Table 2 ijerph-17-05306-t002:** Sleep variables of women in each trimester of pregnancy.

		Study Group *N* = 7207		I Trimester *N* = 791		II Trimester *N* = 2580			III Trimester *N* = 3836		
		No	%	No	%	No	%	*p* I vs. II tr.	No	%	*p* II vs. III tr.
Sleep onset insomnia		1458	20.23	128	16.18	318	12.33	<0.001	1012	26.38	<0.001
Night sleep duration	<4 h	160	2.22	17	2.16	53	2.05	0.92	90	2.34	0.53
	4–6 h	2315	32.12	245	30.97	839	32.52	0.46	1231	32.09	0.62
	7–8 h	3408	47.29	379	47.91	1204	46.67	0.51	1825	47.58	0.59
	>8 h	1324	18.37	150	18.96	484	18.76	0.91	690	17.99	0.84
Too early weakening		1338	18.56	139	17.57	467	18.1	0.82	732	19.08	0.44
Frequent awakening		3803	52.77	411	51.95	980	37.98	<0.001	2412	62.88	<0.001
Sleep medications		125	1.73	18	2.28	54	2.13	0.78	53	1.43	0.06
Too shallow sleep		708	9.82	77	9.73	245	9.5	0.92	386	10.06	0.81
Daytime drowsiness		4975	69.04	683	86.35	1413	54.76	<0.001	2879	77.05	<0.001
Naps impeding daytime function		1686	23.39	215	27.18	568	22.02	0.01	903	23.54	0.37

No-number; h-hours; All analyses: Fisher exact test.

**Table 3 ijerph-17-05306-t003:** Logistic regression analysis of factors influencing the occurrence of sleep problems.

	aOR	95% CI	*p*
Age	1.02	1.00–1.04	0.04
Unemployment	0.81	0.69–0.96	0.03
Stress level	1.12	1.09–1.16	0.001
Assessment of relationship	0.93	0.89–0.98	0.02
Time to conceive 6-12 months	1.31	1.09–1.57	0.03
Nausea	2.11	1.68–2.65	0.001
Backpain	1.94	1.69–2.22	0.001
Nocturnal urination	2.32	2.06–2.63	0.001
Breathing difficulties in supine position	2.59	2.13–3.16	0.001
Leg cramps	1.53	1.33–1.77	0.001
Fetal movements	2.56	2.1–3.13	0.001

aOR: adjusted odds ratio; 95% CI: 95% confident interval.

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
