# Peer review of "Sleep Problems in Pregnancy—A Cross-Sectional Study in over 7000 Pregnant Women in Poland"

_ijerph, 2020, doi:10.3390/ijerph17155306_

Round 1
Reviewer 1 Report
This manuscript characterizes sleep in pregnant women in Poland and analyzes the relation to many factors, including sociodemographic characteristics, pregnancy-related physical symptoms, and sleep problems. The study used self-composed questionnaires for this study. The authors found, not surprisingly, that a significant percentage of women faced sleep problems during pregnancy. The data from this study is interesting and points to counseling women on sleep patterns, however some questions remain that the authors should address prior to publishing this work.
- One of the most significant concerns, is that the investigators did not use a validated survey. Although they mention this, they did not state the reasons for this. There are many surveys that could have been translated for the purpose of this study. The authors need to be more forthcoming why they elected to go this route.
- The exclusion criteria for this study was limited. Thus, those who have sleep disorders were not eliminated from the analysis. The authors need to address this as many of these disorders could be present prior to their pregnancy. This would not alter the data, but may alter the conclusion of how women are counseled as sleep hygiene could be discussed prior to becoming pregnant.
- Many of the self-composed questions are vague and are likely to complicate the results. For example; Do you think your economic status is xxx”? These are questions of perception and should be removed from the analysis in light of the employment data.
- The employment data should be qualified in the limitations since it is unclear if full-time or part-time.
- One question missing is if these women’s employment was on normal schedule or if they did shift-work? This could have significant impacts on this data set, especially as a ~83% were employed. Do the authors have any indication of the work schedules of these women. This would certainly temper their conclusions.
Author Response
Honourable Reviewer,
Thank you for your time and all the valuable suggestions which helped us improve the paper. Here are the answers to your comments:
- One of the most significant concerns, is that the investigators did not use a validated survey. Although they mention this, they did not state the reasons for this. There are many surveys that could have been translated for the purpose of this study. The authors need to be more forthcoming why they elected to go this route.
We are aware of the limitation resulting from using a non-validated survey. We decided to use a self-composed questionnaire in order to join several different factors for further analysis. The questionnaire was attached as a supplementary material. Except of questions concerning demographic and socioeconomical data, we used questions derived from PSQI, Insomnia Severity Index, Stanford Sleep Questionnaire and Berlin Questionnaire to gain information on any sleep problem existing (instead of assessing sleep quality) and analogue visual scale to assess stress and happiness in a current relationship. This survey self-composed of questions coming from several questionnaires allowed us to gain wide range of information.
- The exclusion criteria for this study was limited. Thus, those who have sleep disorders were not eliminated from the analysis. The authors need to address this as many of these disorders could be present prior to their pregnancy. This would not alter the data, but may alter the conclusion of how women are counseled as sleep hygiene could be discussed prior to becoming pregnant.
Thank you very much for pointing out this important issue. It was added to the Material and Methods section. Conclusions were modified accordingly.
- Many of the self-composed questions are vague and are likely to complicate the results. For example; Do you think your economic status is xxx”? These are questions of perception and should be removed from the analysis in light of the employment data.
The questionnaire was added to the supplementary material and is available for readers. According to your suggestion the questions regarding employment data were removed from the analysis.
- The employment data should be qualified in the limitations since it is unclear if full-time or part-time.
Information on this limitation was added to the manuscript.
- One question missing is if these women’s employment was on normal schedule or if they did shift-work? This could have significant impacts on this data set, especially as a ~83% were employed. Do the authors have any indication of the work schedules of these women. This would certainly temper their conclusions.
Thank you very much for pointing out these issues. Unfortunately we do not have information on details of employment. In our questionnaire only “Are you employed?” was included. We have added this information to the study limitations, as you rightly noticed a shift-job could interfere with our results and induce bias.
Reviewer 2 Report
This is a study using a web-based survey of Polish pregnant women regarding subjective sleep quality. Its strength is the large sample size, though it is otherwise not particularly novel. I have a few concerns regarding mostly the methodology.
Major comments:
The term “self-composed” is not very clear. Please describe how the questionnaire was designed.
Also, for questions of the questionnaire that had several possible answers (e.g. Do you have any of the ailments worsening your nighttime sleep?) were the answer choices provided and women had to choose yes / no or were they to write in the answer?
More information is required to better understand the study population and the source population. What websites was the questionnaire available from? Who were publishers of the websites? Who had access to these websites, how were the websites advertised to women etc. ?
Along similar lines, can the authors estimate what proportion of all pregnant Polish women their sample might represent? How representative is their sample or how might it differ from the overall pregnant population. How does self-selection to participate in the study affect results?
What electronic platform was the questionnaire administered from?
“A power analysis was performed to assess the minimal number of respondents. For the 90% power 1000 women should be included in the study.” This is not an adequate sample size calculation description. What outcome was this based on? What statistical test?
For the analyses in table 1, a single comparison test should be performed for proportion of women in each category (e.g. education, employment, etc), such as the Chi squared or Fisher exact. There should not be a separate p values for each individual line in the table. Moreover, it is stated that the Mann-Whitney test (Mann-Whitney U test?) was used but this is a non-parametric test for means/ continuous outcomes, not proportions. Same for table 2.
For table 3, there is no point making this statement: “Variables adjusted in the analysis: age, parity, economic status, employment, stress level, marital status, assessment of relationship, time to conceive, pregnancy ailments”. It’s also incorrect – you are showing them as variables affecting sleep not as possible confounders of the relationship between sleep problems and other predictors.
In the discussion: “It is an interesting finding as excessive sleepiness is a common complaint in the first trimester of pregnancy, however in our study it was reported mostly by women in the third trimester as well. ». Mostly… as well… this is confusing and should be reworded, as the prevalence of this finding is in fact higher in the first trimester.
This is however an interesting observation. Authors might be in a position to use their data to assess predictors of sleepiness in their study population, and for example determine if sleep disturbances are associated with sleepiness or not
“The analyzed data are derived from a self-composed questionnaire, which could be the cause of an inherent bias.” Please explain.
“The question of reliability of the results is a valid concern.” Authors might want to be more specific as to what they mean by that? Admittedly, a non-validated questionnaire cannot be said to be reliable. However, I don’t think that women’s self-reported subjective experiences of sleep are of lesser value than data obtained from laboratory polysomnography, though the two measure very different outcomes.
The discussion lists results of many other similar studies but do little to bring forth the significance of this study and implications of results.
Minor:
Many grammatical or wording errors are present including in the abstract. e.g.: “impending” (“impeding?).
“Mental” work is probably not the best term. “Non-physical” might be preferable.
Author Response
Honourable Reviewer,
Thank you for your time and all the valuable suggestions which helped us improve the paper. Here are the answers to your comments:
- The term “self-composed” is not very clear. Please describe how the questionnaire was designed.
In the study we used a questionnaire composed by the authors themselves and not validated. We are aware of the limitation resulting from using a non-validated survey. We decided to use a self-composed questionnaire in order to join several different factors for further analysis. The questionnaire was attached as a supplementary material. Except of questions concerning demographic and socioeconomical data, we used questions derived from PSQI, Insomnia Severity Index, Stanford Sleep Questionnaire and Berlin Questionnaire to gain information on any sleep problem existing (instead of assessing sleep quality) and analogue visual scale to assess stress and happiness in a current relationship. This survey self-composed of questions coming from several questionnaires allowed us to gain wide spectrum of information. The questionnaire and its references is presented below and is available in the supplementary material.
|
POLISH VERSION OF THE QUESTIONNAIRE |
ENGLISH TRANSLATION |
REFERENCE |
|
Ile ma pani lat? |
How old are you? |
Berlin Questionnaire |
|
Ile ma pani wzrostu? |
What is your height? |
Berlin Questionnaire |
|
Ile ważyła pani przed ciążą? |
How much did you weigh before the current pregnancy? |
Berlin Questionnaire |
|
Ile pani waży obecnie? |
How much do you weigh now? |
Berlin Questionnaire |
|
Ile razy rodziła pani poprzednio? |
How many times did you deliver a baby before? |
|
|
W którym trymestrze jest pani obecnie? |
In which trimester of pregnancy are you currently? |
|
|
Jak długo starała się pani zajść w ciążę? |
For how long did you try to conceive? |
|
|
Jakie ma pani wykształcenie? |
What is your education? |
|
|
Czy pracuje pani zawodowo? (nie pracuję/wykonuję pracę fizyczną/wykonuję pracę umysłową) |
Are you employed? (unemployed/physical work/metal work) |
|
|
Czy ocenia pani swój status ekonomiczny jako: bardzo dobry, dobry, średni, zły? |
Do you think you economic status is: bad/average/good/very good? |
|
|
Czy jest pani zamężna? (singielka/zamężna/w związku) |
Are you married? (single/married/in a relationship) |
|
|
Gdzie pani mieszka? (wieś, małe miasto, duże miasto) |
Where do you live? (countryside/town/city) |
|
|
Jak ocenia pani stres codzienny w skali od 1 do 10? |
How much stress are you under in a scale from 1 to 10? |
Visual Analogue Scale |
|
Czy jest pani zadowolona ze swojego związku (w skali od 1 do 10)? |
Are you happy in your current relationship (in a scale from 1 to 10)? |
Visual Analogue Scale |
|
Ile godzin spała pani przed ciążą? |
For how many hours did you sleep before the pregnancy? |
PSQI |
|
Czy dzieli pani sypialnię z inną osobą? |
Do you share the bedroom with another person? |
PSQI, Insomnia Severity Index
|
|
Czy dzieli pani łózko z inną osobą? |
Do you share the bed with another person? |
PSQI, Insomnia Severity Index
|
|
Czy cierpi pani na dolegliwości pogarszające jakość snu? (nudności, bóle pleców, częste oddawanie moczu, trudności w oddychaniu w pozycji na wznak, skurcze nóg, ruchy płodu) |
Do you have any of the ailments worsening your night-time sleep?(nausea, back pains, frequent nocturnal urination, breathing difficulties in supine position, leg cramps, fetal movements causing sleep problems) |
PSQI |
|
Czy odbywa pani drzemki w ciągu dnia? |
Do you take a nap during daytime? |
Stanford Sleep Questionnaire |
|
Ile drzemek w ciągu dnia? |
How many naps a day? |
Stanford Sleep Questionnaire |
|
Ile godzin dziennie pani śpi? |
For how many hours do you sleep during a day? |
Stanford Sleep Questionnaire |
|
Czy ma pani problemy z: zaśnięciem, wybudzaniem w nocy, zbyt wczesnym wybudzaniem, płytkim snem, trudnościami z obudzeniem się, lękami nocnymi, koszmarami> |
Do you have problems during pregnancy with: problems with falling asleep, nocturnal awakening, to early weakening, shallow sleep, difficulties with waking up, night-time anxiety, nightmares? |
PSQI, Athens insomnia scale |
|
Czy stosuje pani leki nasenne? |
Do you take any medications to sleep? |
PSQI |
|
Czy czuje się pani senna w dzień? |
Do you feel drowsy during a day? |
PSQI, Berlin Questionnaire |
- Also, for questions of the questionnaire that had several possible answers (e.g. Do you have any of the ailments worsening your nighttime sleep?) were the answer choices provided and women had to choose yes / no or were they to write in the answer?
The questionnaire had closed questions – they are presented above.
- More information is required to better understand the study population and the source population. What websites was the questionnaire available from? Who were publishers of the websites? Who had access to these websites, how were the websites advertised to women etc. ?
The survey was dedicated to all pregnant Polish speaking women, regardless of inhabitancy or any other factors. It was distributed by web pages and Facebook groups, which were designed for pregnant women. Web pages written in Polish language provided general medical and social information on pregnancy (https://mamaginekolog.pl, https://mjakmama24.pl). Link to the questionnaire was presented also on Facebook groups bringing together women declaring being pregnant at the time (@mamaginekolog, @zdrowa.ciaza, @superciaza, @prenatalprojekt). Websites and Facebook groups were available for any pregnant women without limits and were not advertised at the time of the survey. Information was added to the manuscript.
- Along similar lines, can the authors estimate what proportion of all pregnant Polish women their sample might represent? How representative is their sample or how might it differ from the overall pregnant population. How does self-selection to participate in the study affect results?
The study was conducted in February and March 2018. No data on haw many women in Poland were pregnant at that time is available. However as 384 426 women delivered in 2018 in Poland, 7855 women included in the study would account for 2% of the whole population. We are not able to determine exactly how much our study group is different from the overall pregnant women population. On one hand a large cohort of respondents allows to assume that a wide range of pregnant women were included, while on the other hand online distribution of the questionnaire selects women using these kinds of media. This limitation was added to the manuscript.
- What electronic platform was the questionnaire administered from?
The questionnaire was administered by Google Forms. This information was added to the manuscript.
- “A power analysis was performed to assess the minimal number of respondents. For the 90% power 1000 women should be included in the study.” This is not an adequate sample size calculation description. What outcome was this based on? What statistical test?
More detailed information was added to the manuscript.
- For the analyses in table 1, a single comparison test should be performed for proportion of women in each category (e.g. education, employment, etc), such as the Chi squared or Fisher exact. There should not be a separate p values for each individual line in the table. Moreover, it is stated that the Mann-Whitney test (Mann-Whitney U test?) was used but this is a non-parametric test for means/ continuous outcomes, not proportions. Same for table 2.
The analysed categories were divided to several subgroups and Fisher exact test enables a separate analysis of each one. Chi squared test was not used in our statistical analysis. We believe more specific analysis is much more interesting. We believe that a detailed comparison of the proportions between 2 categories is more favourable in statistical inference than 1 overall value of statistics for a 2x3 or 2x4 table as it reveals more specific differences. The Mann-Whitney test was used for means comparison like age or BMI. Specification of which tests were used is presented in the table legend.
- For table 3, there is no point making this statement: “Variables adjusted in the analysis: age, parity, economic status, employment, stress level, marital status, assessment of relationship, time to conceive, pregnancy ailments”. It’s also incorrect – you are showing them as variables affecting sleep not as possible confounders of the relationship between sleep problems and other predictors.
The statement was remowed.
- In the discussion: “It is an interesting finding as excessive sleepiness is a common complaint in the first trimester of pregnancy, however in our study it was reported mostly by women in the third trimester as well. ». Mostly… as well… this is confusing and should be reworded, as the prevalence of this finding is in fact higher in the first trimester.
Thank you for pointing out this mistake. The sentence was corrected.
- This is however an interesting observation. Authors might be in a position to use their data to assess predictors of sleepiness in their study population, and for example determine if sleep disturbances are associated with sleepiness or not
Thank you for pointing out this issue, which helped us improve the manuscript. This analysis for added to the manuscript.
- “The analyzed data are derived from a self-composed questionnaire, which could be the cause of an inherent bias.” Please explain.
As we decided to use a self-composed questionnaire which was not validated it could induce bias. Specific construction of the questions, like “Are you employed?” without details on a full-time or a part-time job or a night-shift work could induce bias to our analysis as well.
- “The question of reliability of the results is a valid concern.” Authors might want to be more specific as to what they mean by that? Admittedly, a non-validated questionnaire cannot be said to be reliable. However, I don’t think that women’s self-reported subjective experiences of sleep are of lesser value than data obtained from laboratory polysomnography, though the two measure very different outcomes.
“The question of reliability of the results is a valid concern.” – by that we meant rather if women would not cheat about sleep medications usage for example. As it is a survey it is not possible to verify such information.
- The discussion lists results of many other similar studies but do little to bring forth the significance of this study and implications of results.
Thank you for this remark. Emphasis of the significance of the study was added to the manuscript.
Minor:
- Many grammatical or wording errors are present including in the abstract. e.g.: “impending” (“impeding?).
The corrections were made.
- “Mental” work is probably not the best term. “Non-physical” might be preferable.
The corrections were made.
Round 2
Reviewer 2 Report
The authors have made some minor improvement but did not address several comments adequately.
-Regarding the questionnaire, though I understand the authors created their own questionnaire by amalgamating a variety of validated questionnaires, the term “self-composed” remains. This is neither grammatically correct nor informative.
-possible answers to the non-yes or no questions of the questionnaire are not further described.
-who created the Facebook groups where this questionnaire was distributed is still not described in the paper. Readers will not have access to the authors response to reviews.
-the information about the proportion of respondents in relation to the number of women who delivered is interesting and should be included in the manuscript.
-the power calculation is unfortunately no clearer. What exactly was the null hypothesis? Which outcome was the study powered on? What comparison? This is key but not provided. Since the stated objective was to “characterize sleep patterns”, it is not apparent what outcome authors were making power calculations on.
-with respect to response 11 regarding the potential bias related to the questionnaire, authors focus on employment history, but the sleep questions are more of a concern.
-Moreover, this text “Despite its obvious limitations as a research tool” – if the limitations are so obvious to the authors, they should be made explicit for the reader.
Author Response
Honourable Reviewer,
Thank you for your time and all the valuable suggestions which helped us improve the paper. We hope that after inducing this changes the manuscript will satisfy your expectations. Here are the answers to your comments:
- Regarding the questionnaire, though I understand the authors created their own questionnaire by amalgamating a variety of validated questionnaires, the term “self-composed” remains. This is neither grammatically correct nor informative.
Yes, we created the questionnaire by combination of different questions derived from several validated questionnaire. This information was added to the manuscript.
- possible answers to the non-yes or no questions of the questionnaire are not further described.
All questions were closed. The non-yes or no questions concerned age, height, weight, parity, current trimester of pregnancy, time to conceive (<6 months, >6 months, >1 year, ART), education (primary, vocational , secondary, higher), employment (unemployed/physical work/metal work), marital status (single/married/in a relationship), inhabitancy (countryside/town/city), pregnancy ailments (nausea, vomiting, back pains, frequent nocturnal urination, breathing difficulties in supine position, leg cramps, fetal movements causing sleep problems), sleep problems (problems with falling asleep, nocturnal awakening, to early weakening, shallow sleep, difficulties with waking up, night-time anxiety, nightmares), stress and happiness in a relationship in a scale from 1 to 10. The questionnaire is presented in the supplementary materials and answers with their percentages are presented in results section.
- who created the Facebook groups where this questionnaire was distributed is still not described in the paper. Readers will not have access to the authors response to reviews.
The facebook groups which shared the questionnaire created before the survey. The asked their administrators for help I sharing a link to the questionnaire and the consent was given to do so. @mamaginekolog was created by mamaginekolog, @prenatalprojekt by Sebastian Kwiatkowski. We do not know the names of @zdrowa.ciaza and @superciaza founders.
- the information about the proportion of respondents in relation to the number of women who delivered is interesting and should be included in the manuscript.
Thank you very much for this suggestion. The information was added to the text.
- the power calculation is unfortunately no clearer. What exactly was the null hypothesis? Which outcome was the study powered on? What comparison? This is key but not provided. Since the stated objective was to “characterize sleep patterns”, it is not apparent what outcome authors were making power calculations on.
We used a single proportion test. The value of the zero hypothesis was established at 90%, as basing on the available published polish data we assumed that 90% of pregnant women declare having any sleep problems. This information was added to the manuscript.
- with respect to response 11 regarding the potential bias related to the questionnaire, authors focus on employment history, but the sleep questions are more of a concern.
We are aware of the possible bias related to the questionnaire. However we asked about occurrence of listed sleep problems in our respondents subjective opinion, without using any objective tools like polysomniography. The aim of our study was to characterize sleep patterns in pregnant women in Poland according to their subjective opinion and the analysis for performed in this regard.
- Moreover, this text “Despite its obvious limitations as a research tool” – if the limitations are so obvious to the authors, they should be made explicit for the reader.
In our opinion obvious limitations of the questionnaire is impossibility to obtain verifiable information, as the survey enables to collects only objective opinions of the respondents and cannot rule out lies. This information was added to the manuscript.
This manuscript is a resubmission of an earlier submission. The following is a list of the peer review reports and author responses from that submission.
Round 1
Reviewer 1 Report
The authors must be acknowledged for the enomous work they carried out, a work never performed before in Poland. Unfortunately the manuscript suffers serious flaws
First, there is a control group of pregnant women with no sleep problem but no control group of non pregnant women. This is a severe limitation for the interpretation of the data 1. Introduction The authors point out that the most common sleep-related disorders during pregnancy include insomnia, restless legs syndrome, obstructive sleep apnea syndrome and night-time gastro-oesophageal reflux disease. Unfortunately, they refer only to insomnia in their manuscript. Indeed, according to the literature the occurrence of insomnia in the Polish population of pregnant women was estimated at 25-40% while overall sleep disorders rate was estimated at 84.2-90.5%. Thus, it is surprising the paper only deals with insomnia 2. Material and methods The number of recruited pregnant women is not indicated in the presentation of the materials The whole study is based on a self-composed questionnaire containing questions in the Polish language. Apparently this questionnaire has not yet been validated. The same applies to the visual numerical scale from 0 to 10 used to assess both the life stress level and the satisfaction with the current relationship Thus, the study relies exclusively on an unknown subjective test, while, in 2020, it should rely on objective and subjective validated tests: an objective test such as a wrist actigraphy continuously worn for 7 days, a well validated and non cumbersome tool to measure sleep duration and subjective tests such as a Pittsburgh sleep quality index (PSQI) to assess the self reported sleep quality, a depression scale and a sleepiness scale The first part of the questionnaire included questions regarding sociodemographic data and information on the current pregnancy, including pregnancy ailments. The sociodemographic data are presented on table 1 but the information on the current pregnancy is not disclosed. The second part contained questions regarding sleep patterns over the past four weeks, but the results bear on 9 or 3 months and it is not clear how the authors used sleep patterns over the past four weeks spanning 2 terms 3. Results 7855 pregnant women took part in the survey. This is an enormous sample. Yet, it is not clear whether the authors conducted a test of statistical power granted to calculate the number of necessary subjects to get significant results Figure 1. The percentage of pregnant women sleeping 7 hours or longer declined from the first to the third term while the number of those sleeping 4 to 6 hours per night increased with pregnancy. This is a potentially interesting finding. However, it is not clear why there is a gap between 6 and 7, and an overlap between 7 and 8, and 7 or longer The study group was further divided into women declaring having or not at least one of the sleep problems listed in the questionnaire. Sleep problems were reported each term, fine; but main problems (sleep onser insomnia, awaking too early, sleep too shallow) weren't reported each term Why is that? 4. Discussion Paragraph 1: According to the Brazilian study pregnant women more commonly suffered from excessive daytime sleepiness with a highest rate of 55% in the second term This result is unexpected as most studies indicate excessive sleepiness as a common first-term complaint. Paragraph 3: Several authors insist on the role of depression in the development of insomnia during pregnancy. It is regrettable that this role was not investigated in this study Strength and Limitations The uniquely large cohort of pregnant women would be a definite strength of the study if this number had been statistically calculated, which does not seem to be the case The lack of validated objective and subjective tests is another limitation of the study
Author Response
Honourable Reviewer,
Thank you for your time, engagement and all the suggestions to improve the quality of our paper. Here are the answers to your comments:
- First, there is a control group of pregnant women with no sleep problem but no control group of non pregnant women.
Only women pregnant at the time of the survey were included in the study. Unfortunately for the reason the control group of non-pregnant women were not available for analysis. This has been added to the limitations of the study.
- The authors point out that the most common sleep-related disorders during pregnancy include insomnia, restless legs syndrome, obstructive sleep apnea syndrome and night-time gastro-oesophageal reflux disease. Unfortunately, they refer only to insomnia in their manuscript. Indeed, according to the literature the occurrence of insomnia in the Polish population of pregnant women was estimated at 25-40% while overall sleep disorders rate was estimated at 84.2-90.5%. Thus, it is surprising the paper only deals with insomnia.
We have decided to focus on sleep problems which can be easily assessed by a questionnaire distributed via the internet as we have chosen this method of a survey to reach as many respondents as possible. Therefore only subjective sleep problems could be analysed. Medical conditions like restless legs syndrome, obstructive sleep apnea syndrome or night-time gastro-oesophageal reflux disease are difficult or impossible to diagnose only on the basis of the questionnaire and could induce bias into the study.
- The number of recruited pregnant women is not indicated in the presentation of the materials
The number of respondents was given in the results section. It has been added to the materials as well.
- The whole study is based on a self-composed questionnaire containing questions in the Polish language. Apparently this questionnaire has not yet been validated. The same applies to the visual numerical scale from 0 to 10 used to assess both the life stress level and the satisfaction with the current relationship.
Thank you for the comment. Indeed the questionnaire was composed by the authors themselves and it was not validated. This information was added to the paper. For the assessment of life stress level and satisfaction with relationship visual analogue scale was used. VAS scale is a validated instrument.
- Thus, the study relies exclusively on an unknown subjective test, while, in 2020, it should rely on objective and subjective validated tests: an objective test such as a wrist actigraphy continuously worn for 7 days, a well validated and non cumbersome tool to measure sleep duration and subjective tests such as a Pittsburgh sleep quality index (PSQI) to assess the self reported sleep quality, a depression scale and a sleepiness scale
We have conducted a survey based on a questionnaire distributed online to reach a large and diverse group of people all over the country. We would not be able to perform any objective sleep measurements in over 7000 women. However we did use PSQI in another survey conducted in the same group of women – the results are analysed and will be published in the future.
- The first part of the questionnaire included questions regarding sociodemographic data and information on the current pregnancy, including pregnancy ailments. The sociodemographic data are presented on table 1 but the information on the current pregnancy is not disclosed.
The analysed information on the current pregnancy included primiparity/multiparity, current trimester of pregnancy and pregnancy ailments. They are presented in table 1 and 2. As all data were available only through a questionnaire, we did not ask about specific pregnancy complications, as respondents’ subjective opinion of having or not specific medical conditions were not reliable in our opinion and could induce bias.
- The second part contained questions regarding sleep patterns over the past four weeks, but the results bear on 9 or 3 months and it is not clear how the authors used sleep patterns over the past four weeks spanning 2 terms
Only women pregnant at the time of the survey were included in the study and were asked to answer the questions regarding the past for weeks before completing the questionnaire. Women being currently in the first, second and third trimester were then analysed separately.
- Yet, it is not clear whether the authors conducted a test of statistical power granted to calculate the number of necessary subjects to get significant results
Before the study was performed we have conducted a power analysis to calculate the minimal number of respondents. It was found that 1000 respondents would be necessary to gest significant results. As the response to our survey was so huge, we decided to include all women who filled in the questionnaire completely. It has been added to the manuscript.
- Figure 1. The percentage of pregnant women sleeping 7 hours or longer declined from the first to the third term while the number of those sleeping 4 to 6 hours per night increased with pregnancy. This is a potentially interesting finding. However, it is not clear why there is a gap between 6 and 7, and an overlap between 7 and 8, and 7 or longer
Thank you very much for the comment as there is a mistake in the manuscript. The correct time intervals should be: <4, 4-6, 7-8, >8. It has been corrected in the paper. The respondents were asked to specify their mean sleep time during the last four weeks. They were asked to round the time to the whole hours, like 4 hours, 5 hours etc. As the rates of women sleeping 7-8 and >8 hours were similar in each trimester, they were additionally analysed jointly (data presented in the text).
- Sleep problems were reported each term, fine; but main problems (sleep onser insomnia, awaking too early, sleep too shallow) weren't reported each term Why is that?
The questionnaire included specific questions regarding all analysed sleep problems, but for clear and easy statistical analysis they were all analysed jointly.
- Paragraph 1: According to the Brazilian study pregnant women more commonly suffered from excessive daytime sleepiness with a highest rate of 55% in the second term This result is unexpected as most studies indicate excessive sleepiness as a common first-term complaint.
Thank you for this interesting remark. It has been added to the manuscript.
- Paragraph 3: Several authors insist on the role of depression in the development of insomnia during pregnancy. It is regrettable that this role was not investigated in this study
Thank you very much for this comment. We are very sorry as the role of depression in the development of sleep problems was not investigated in our study.
- The lack of validated objective and subjective tests is another limitation of the study
It has been added to the manuscript.
Reviewer 2 Report
Thank you for this study on sleep problems in Polish-speaking pregnant women. This is an important topic of research at present and it is interesting to have data from larger pools of pregnant women from around the world. I note that there are some limitations: the survey was distributed online to a convenience sample and this, together with associated response bias, is a significant limitation of the study when it comes to the estimation of prevalence. Another limitation is the inclusion of singleton sample only, as most of the research has been done on singleton pregnancies, with multiple pregnancies being excluded. If the authors wish to run a similar study in future, it may be worthwhile targeting multiple pregnancies specifically. Self-reported data is a limitation, as noted by the authors.
MAJOR COMMENTS
The main issues that can be improved in the manuscript are the statistical analyses and the description of the methods and results.
Methods: More detail is required, particularly on participant recruitment, which influences what types of people are recruited into the study.
- It sounds like participants were recruited online via Facebook and other websites. Can the authors please provide examples of the type of websites involved? Are these websites providing general information about pregnancy? Are they government websites or hospital websites?
- Can the authors please provide information on how the survey was advertised to participants i.e. was the intent of the survey to look at sleep in pregnancy clear to participants? Or was the advertising generally based around “pregnancy”?
- I am unclear as to what “self-composed” questionnaire means.
- The Supplementary Material is not a true reflection of the original survey, which was in Polish. The Methods refer to “sleep patterns over the past 4 weeks” and the description sounds a lot like the Pittsburgh Sleep Quality Index. Can the authors please confirm if any standardised questionnaires were included in the survey?
Results
- Whilst the data is valuable, the current statistical analyses of the results are inappropriate –logistic regression should include the entire categorical variable and not just dummy/indicator variables of each category. Categorical variables should be analysed as categorical and these categories preserved in the logistic regression.
- Table 1 should include the test statistic for each comparison, not just the p-value. For categorical variables from Education to Time to Conceive, the appropriate test statistic should be the chi-squared test (or Mantel Hanzel chi-squared) and there should only be one result for each variable. Pair-wise comparisons across each subcategory of a variable is not appropriate.
- It would be worthwhile including Trimester in Table 1.
- Please include descriptive statistics for the symptom variables in Table 1.
- I suggest a Table 2 in which sleep variables are described by Trimester (similar to Figure 1, but detailing all of the sleep variables, not just sleep duration).
- In the Results text, it is unclear what statistical test was carried out for the comparisons for sleep duration by trimester. A better way of showing this comparison is to include 95% CI on the bar chart (Figure 1).
- Table 2 results should be consistently presented to 2 decimal places. It is good practice to include the p-values as well as the 95% CI in the table.
MINOR COMMENTS
Title
- Should include the study population in the title i.e. “Polish women with singleton pregnancies”
Abstract
- Abstract refers to a “self-composed” questionnaire. Do you mean “self-reported”?
- It would be helpful to include some mention of the target population/how participants were recruited in the abstract, to understand how the convenience sample was derived.
- Suggest deletion of the sentence “The analysed demographic, socioeconomic and psychological issues had a 19 small but significant influence on sleep problems occurrence (aOR 0.81–1.24)” and instead mention how these variables were adjusted for in the results that follow.
- “Time to conceive from 6 to 12 months of trying increased the risk of developing sleep problems during pregnancy (aOR 1.31). Pregnancy ailments increased the risk of sleep disturbances (aOR 1.53–2.59). Sleep disorders are prevalent among pregnant women in Poland.” These results are stated too strongly – this is an observational study and these statements indicate causality which cannot be shown through observation. Similarly, “sleep disorders” is a strong claim when self-reported symptoms are described. Suggest using the words “associated with increased risk” and “sleep problems are prevalent” instead.
Introduction
- Line 44 – “od” should be “of”
Author Response
Honourable Reviewer,
Thank you for your time and all the valuable suggestions which helped us improve the paper. Here are the answers to your comments:
- Methods: More detail is required, particularly on participant recruitment, which influences what types of people are recruited into the study.
Thank you very much for this suggestion. We have described in more detail the recruitment od the respondents and added it to the methods section.
It sounds like participants were recruited online via Facebook and other websites. Can the authors please provide examples of the type of websites involved? Are these websites providing general information about pregnancy? Are they government websites or hospital websites?
Only social web pages providing general medical and social information on pregnancy were used (e.g. prenatalprojekt.pl). No government or hospital websites were used.
Can the authors please provide information on how the survey was advertised to participants i.e. was the intent of the survey to look at sleep in pregnancy clear to participants? Or was the advertising generally based around “pregnancy”?
On websites and Facebook group a short information on sleep problems in pregnancy and its importance was presented with with an invitation to complete the survey. This information was added to the manuscript.
- I am unclear as to what “self-composed” questionnaire means.
By a self-composed questionnaire we meant that we have written the questions by ourselves without copying any other known questionnaire.
The Supplementary Material is not a true reflection of the original survey, which was in Polish. The Methods refer to “sleep patterns over the past 4 weeks” and the description sounds a lot like the Pittsburgh Sleep Quality Index. Can the authors please confirm if any standardised questionnaires were included in the survey?
In Supplementary Material a questionnaire originally written in Polish was translated into English so that it would be available to people who don’t speak Polish. We decided to use the period of the last 4 weeks before completing the questionnaire because in another part of this survey we did use PSQI. The second part is being analysed and we hope to publish it in another paper.
- logistic regression should include the entire categorical variable and not just dummy/indicator variables of each category. Categorical variables should be analysed as categorical and these categories preserved in the logistic regression.
We have conducted logistic regression analysis and firstly we have included all qualitative and quantitative data. In Table 2 only those factors who were found independent risk factors were presented.
- Table 1 should include the test statistic for each comparison, not just the p-value. For categorical variables from Education to Time to Conceive, the appropriate test statistic should be the chi-squared test (or Mantel Hanzel chi-squared) and there should only be one result for each variable. Pair-wise comparisons across each subcategory of a variable is not appropriate.
The Mann-Whitney test was used to compare quantitative and Fisher exact tests was used to compare the categorical variables. Test statistic information was added to table 1.
- It would be worthwhile including Trimester in Table 1.
Thank you very much for your valuable suggestion. It has been added to table 1.
- Please include descriptive statistics for the symptom variables in Table 1.
The descriptive statistics has been added in the table legend.
- I suggest a Table 2 in which sleep variables are described by Trimester (similar to Figure 1, but detailing all of the sleep variables, not just sleep duration). In the Results text, it is unclear what statistical test was carried out for the comparisons for sleep duration by trimester. A better way of showing this comparison is to include 95% CI on the bar chart (Figure 1).
Thank you very much for the suggestion. We have added a separate table presenting sleep variables in each trimester. It was labelled table 2 and the previous table 2 to maned table 3.
- Table 2 results should be consistently presented to 2 decimal places. It is good practice to include the p-values as well as the 95% CI in the table.
All results were changed to 2 decimal places. p values were added to the table.
- Should include the study population in the title i.e. “Polish women with singleton pregnancies
As Polish speaking women may not be Polish women, we changed the title to “Sleep problems in pregnancy – a cross-sectional study in over 7000 pregnant women in Poland
- Abstract refers to a “self-composed” questionnaire. Do you mean “self-reported”?
By a self-composed questionnaire we meant that we have written the questions by ourselves without copying any other known questionnaire
- It would be helpful to include some mention of the target population/how participants were recruited in the abstract, to understand how the convenience sample was derived.
Short information was added to abstract.
- Suggest deletion of the sentence “The analysed demographic, socioeconomic and psychological issues had a 19 small but significant influence on sleep problems occurrence (aOR 0.81–1.24)” and instead mention how these variables were adjusted for in the results that follow.
Suggested changes were made in the paper.
- Sleep disorders are prevalent among pregnant women in Poland.” These results are stated too strongly – this is an observational study and these statements indicate causality which cannot be shown through observation. Similarly, “sleep disorders” is a strong claim when self-reported symptoms are described. Suggest using the words “associated with increased risk” and “sleep problems are prevalent” instead
Suggested changes were made in the paper.
- Line 44 – “od” should be “of”
Suggested change was made in the paper.
Round 2
Reviewer 1 Report
The authors have improved the quality of their work in suppressing figure 1, completing table 1 (Pregancy trimester) and adding table 2 (Sleep variables of women in each trimester of pregnancy)
Unfortunately they could not correct major flaws of their manuscript, that is to say
- Absence of a control group of non pregnant women - Use of a non validated self composed questionnaire - Absence of objective measure of sleep duration - Absence of a validated subjective test to assess sleep quality - Absence of a depression scale
- No information on sleep problems apart from insomnia (restless legs syndrome, obstructive sleep apnea syndrome, night-time gastro-oesophageal reflux)
Author Response
Thank you very much for your time and valuable comments that would make our manuscript better. However, despite our most sincere intentions, we are not able to make further corrections of the manuscript. We conducted the survey in a group of pregnant women 2 years ago and no additional group of non-pregnant women was included that time. In our survey the study group was the group of pregnant women declaring having sleep problems and control group was the group declaring no sleep problems during gestation. It is not possible for us to add another group of non-pregnant women now without conducting the whole survey again from the beginning.
In the study we used a questionnaire composed by the authors themselves and not validated. This information was added to the paper. We are not able to change it now. We intend to conduct and publish another study based on the Pittsburgh Sleep Quality Index in the future.
As our study was conducted via online questionnaire only subjective opinion on sleep duration could be collected. We aimed to reach a large and diverse group of pregnant women all over the country. We would not be able to perform any objective sleep measurements in over 7000 women and therefore we decided to base on information provided by respondents. We are aware of the bias it could introduce and it was discussed as the limitation of our study. It was not possible to gain objective information on restless legs syndrome, obstructive sleep apnea syndrome or night-time gastro-oesophageal reflux via a questionnaire as well.
The aim of our study was to characterise sleep patterns of pregnant women according to their subjective opinion. We did not analyse the occurrence of depression or depressive symptoms. We are very sorry as the role of depression in the development of sleep problems was not investigated in our study.